# Enhancement of diterpenoid steviol glycosides by co-overexpressing *SrKO* and *SrUGT76G1* genes in *Stevia rebaudiana* Bertoni

Nazima Nasrullah[1]*, Javed Ahmad[1], Monica Saifi[1], Irum Gul Shah[2], Umara Nissar[3], Syed Naved Quadri[1], Kudsiya Ashrafi[1], Malik Zainul Abdin[1]*

**1** Department of Biotechnology, CTPD, School of Chemical and Lifesciences, Jamia Hamdard, New Delhi, India, **2** Division of Genetics, IARI- Indian Agricultural Research Institute, New Delhi, India, **3** Department of Biomedical Research, University of Bern, Bern, Switzerland

* mzabdin@jamiahamdard.ac.in (MZA); nazima.tajali@gmail.com (NN)

## Abstract

*Stevia rebaudiana* (stevia) contains commercially important steviol glycosides, stevioside and rebaudioside A, these compounds have insulinotropic and anti-hyperglycemic effect. Steviol, stevioside and rebaudioside-A have taste modulation and insulin potentiation activity. Stevia leaves are composed of steviol (2–5%), stevioside (4–13%) and rebaudioside-A (1–6%). Stevioside has after-taste bitterness, rebaudioside-A is sweetest in taste among all the glycosides present. Therefore, lower ratio of rebaudioside-A to stevioside has bitter after-taste, which makes stevia plants unpalatable. By over-expressing the genes, *SrUGT76G1* and *SrKO*, we propose to increase the ratio of RebA to stevioside in stevia. Various lines were generated and amongst them, seven lines had both the transgenes present. Co-overxpresion of *SrUGT76G1* and *SrKO* led to the increased concentration of RebA in all the seven transgenic lines (KU1-KU7) than control plant and RebA to stevioside ratio also increased significantly. Steviol, stevioside and RebA showed a differential concentration in all the seven lines, but the pattern was the same in all of them and the ratio of RebA to stevioside increased dramatically. In transgenic line 2 (KU2), RebA showed a steep increase in concentration 52% the rebaudioside-A to stevioside ratio increased from 0.74 (control) to 2.83. In overall all the lines, RebA showed a positive correlation with steviol and stevioside. Overexpression of *SrKO* led to an increase in steviol which increased the stevioside, overexpression of *SrUGT76G1* ultimately increased RebA concentration. In conclusion, concentration of RebA increased significantly with co- overexpression of *SrUGT6G1* and *SrKO* genes. Lines with increased RebA are more palatable and commercially viable.

## 1. Introduction

Among the natural sweeteners reported so far, steviol glycosides (SGs) are 200–300 times sweeter than sucrose [1]. They are present in *Stevia rebaudiana* Bertoni (Stevia), a perennial herb from the family Asteraceae native to eastern Paraguay, Barazil. Typically, these compounds include

**Data Availability Statement:** The GC-MS file is available through github (https://github.com/naizma/GC-MS-STEVIA-REBAUDIANA). This is the

raw GC-MS data from which we generated Heatmap, correlation table and other pics in the online web based interface MetaboAnalyst. Original gel images are uploaded as S1 Raw Images.

**Funding:** This study was funded by DSR(SAP)-University grants commission New Delhi-India to Professor M.Z.Adin. The funders had no role in study design, data collection and analysis, decision to publish, or preparation of the manuscript.

**Competing interests:** The authors have declared that no competing interests exist.

stevioside (4–13% dry weight), steviolbioside (trace), the rebaudiosides, including rebaudioside A (RebA) (1–6%), rebaudioside B (trace), rebaudioside C (1–2%), rebaudioside D (trace), rebaudioside E (trace), rebaudioside F (trace) and dulcoside A (0.4–0.7%) [2–4].

After final nod from the food safety authorities around the world, steviol glycosides are being used in commercial beverages, confectionaries and foods worldwide. Demand of stevia is hence increasing day by day. SGs modulate taste responses and insulin release by activating TRPM5 (Transient receptor potential cation channel subfamily M member 5) ion channel, and the potentiation of TRPM5 activity protects mice against the development of high-fat diet induced hyperglycaemia. Thus stevioside, RebA and their aglycon steviol are novel contenders for the development of antidiabetic drugs [5].

Methylerythritol Phosphate Pathway (MEP) and the Mevalonate Pathway (MVA) pathways are the principal contributors of sesquiterpenes, whereas mono- and diterpenes, in particular SGs, are exclusively formed by the MEP pathway [6]. Like all diterpenes, steviol is synthesized from geranylgeranyl diphosphate (GGDP). The step in gibberellic acid (GA) biosynthesis is catalyzed by ent-kaurene oxidase (*KO)* leading to the synthesis of ent-kaurenoic acid from ent-kaurene before the branch point to steviol production [7]. Ent-kaurene oxidase in *Arabidopsis*, *Cucurbita maxima* and pea are expressed in developing seeds and young, rapidly growing tissues with low expression levels in mature leaves. In stevia, however, Ent-kaurene oxidase (*SrKO*) expresses in both young and mature leaf tissues [8]. Stevia *SrKO* is highly expressed in leaves, succulent stems, flowers and seedling shoots. Gene expression of *SrKO* has a positive correlation with the concentration of stevioside in stevia leaves [9].

UDP-glucosyltransferase 76G1 (*SrUGT76G1*) catalyses the conversion of stevioside to RebA via one-step glycosylation reaction, which increases the amount of sweet-tasting RebA and decreases the amount of stevioside that has a bitter aftertaste [10]. *SrUGT76G1* has glycosylating activity, not only towards aliphatic and branched alcohols but also towards (substituted) phenols, flavonoids, gallates and glycosides. *SrUGT76G1* have remarkably broad specificity, which can be exploited for the glucosylation of a wide range of acceptor molecules [11]. UDP-glucosyltransferases are not highly specific and show broad substrate specificity based on a regioselectivity. There is a correlation between the higher SGs accumulation and the transcription of *SrUGT76G1* [12].

Guleria and Yadav in 2013 through RNAi technology downregulated the key genes, *SrKA13H*, *SrUGT85C2*, *SrUGT74G1* and *SrUGT76G1* of steviol glycoside biosynthetic pathway. They found a 60% reduction in overall steviol glycoside content. Base mutations in PSPG box and amino acid substitution mutations, which changed the secondary and tertiary structure of protein *SrUGT76G1*, have reduced the RebA content in stevia plant [13].

So, we designed our research to develop a novel variety of sweet herb, stevia with a higher ratio of RebA to stevioside in the leaves that may result in the loss of liquorice like aftertaste. We used metabolic engineering approach to alter the biosynthetic pathway of SGs in stevia plant, leading to the improved biosynthesis of steviol glycosides and increased ratio of RebA to stevioside. In the present study, we over expressed ent-kaurene 19-oxidase and UDP-glycosyl transferase 76 G1 genes in stevia and evaluate their influence on the plant growth characteristics, stevioside and RebA contents their yield and ratio.

## 2. Materials and methods

### 2.1 Chemicals and reagents

Chemicals for molecular research were purchased from Sigma Aldrich (India), Roche (USA) and Promega Life Science (India). Restriction enzymes and plasmid isolation kits were obtained from Thermo Fisher Scientific, Germany. Gel Elution, RNeasy and DNAeasy Plant

Mini Kits were purchased from Qiagen, Germany. Tissue culture grade chemicals were obtained from Himedia (Mumbai, India) and Merck (India). Glass wares were obtained from Schott Duran and Borosil (India). Disposables plastic wares were obtained from Axygen and Tarsons (India). Methanol, hexane, ethanol, chloroform and acetonitrile were of HPLC grade and procured from Merck (India). The standard samples of stevioside and RebA were of analytical grade and purchased from Sigma-Aldrich (USA). Silica plates for HPTLC were purchased from Merck (India).

## 2.2 Plant material

Stevia (CIM-Mithi) plants were obtained from Central Institute of Medicinal and Aromatic Plants (CIMAP), Pant Nagar, Uttrakhand, India. The plants were grown in plastic pots in equal ratio of soil and sand in poly house at Jamia Hamdard, New Delhi, India under controlled conditions, in an artificial climate chamber programmed at 25˚C/16˚C 16 h/8 h day/night environment at relative humidity, 60%, with a light intensity of 300 $\mu mol/m^2 s^{-1}$.

## 2.3 Cloning of genes in plant binary vector

RNA was isolated from leaf sample (100 mg) by RNeasy Plant Mini Kit (Qiagen) following the suppliers' instructions. The DNA impurity was removed by incubating RNA samples with RNase free DNase I (Sigma-Aldrich, USA). The concentration and purity of RNA samples were measured by NanoDrop spectrophotometer (ND1000). The stability of RNA samples were evaluated by subjecting it to 1.5% agarose gel electrophoresis. cDNA was synthesized by Verso cDNA Kit following the suppliers instructions. The transcript of *SrUGT76G1* (FP: `ATG GAAAATAAAACGGAGAC` RP: `TTACAACGATGAAATGTAAG`) and *SrKO* (FP: `ATGGATGCCGT CACCGGTTTGCTG` RP: `TCATATCCTGGGCTTTATTATGGCC`) were amplified by using their gene-specific primers having BamHI at 5´ end and SacI at 3´ end. After confirmation by sequencing, transcripts with desired restriction sites were cloned between BamHI and SacI sites in plant binary vector pBI121 having the size of 14,758bp and *GFP* gene as reporter gene. Recombinant pBI121 vector was then mobilized into *Agrobacterium tumefacians* EHA105.

## 2.4 *Agrobacterium* transformation screening

Transformed *A. tumefacians* EHA105 was streaked on LB plate with antibiotics rifampicin (15mg/L) and kanamycin (20mg/L) and was grown for 2 days at 28˚C. Single colony from the plate was added to 5ml tube of LB and grown overnight at 28˚C on a rotating shaker. 500 μl (1:100 dilution) of culture from the tube was added to 50 ml LB in a 125 ml Erlenmyer flask and incubated overnight at 28˚C on a rotating shaker with appropriate antibiotics. Culture (1.5mL) having absorbance of 1.0 AU at 600 nm was pelleted down at 1,000 rpm for 10 min at room temperature (24˚C), supernatant was discarded, pellet was re-suspended in infiltration medium (250mg D-glucose, 5ml of 500mM MES, 5ml of 20mM $Na_3PO_4.12H_2O$ and 5μl of 1M acetosyringone) (Table 1) to get desired density ($OD_{600}$ of 0.8). Culture of 1ml of

**Table 1. Composition of reagents to make 50 mL of infiltration medium.**

| Infiltration medium (to make 50 ml) | |
|---|---|
| 500mM MES (Sigma) | 4.88 g in 50 ml made up with $dH_2O$. Store at 4 ˚C |
| 20mM $Na_3PO_4$ $12H_2O$ (trisodium orthophosphate; BDH) | 0.38 g in 50 ml made up with $dH_2O$. Store at 4 ˚C. |
| 1 M acetosyringone (3',5'-dimethoxy-4'-hydroxy acetophenone; Aldrich) | 0.196 g in 1 ml made up with DMSO. Divide into single use aliquots and store at–20 ˚C |

*Agrobacterium* harboring *SrKO* and *SrUGT76G1* cDNA were used for individual transformation. For co-expression, cultures of *Agrobacterium* harboring *SrKO* and *SrUGT76G1* were mixed in the ratio of 1:1, respectively. Leaves of five-month-old plants were gently rubbed and infiltrated with a hypodermic syringe (needle removed) [14].

## 2.5 Transient transformation

Healthy plants were removed from the growth chamber and placed under a white fluorescent lamp for 1 h before infiltration to open the stomata fully to aid infiltration. Third or fourth leaves of plants from the apical meristem were chosen. Multiple leaves on the same plant were infiltrated in midrib regions or in other independent plants. Leaves to be infiltrated were prepared by gently rubbing a small region (approximately 0.5 cm$^2$) of the underside of the leaf to remove the wax cuticle. Before infiltration, leaves were marked for ease of identification. Resuspended *A. tumefacians* cells in a 1-ml syringe (needle removed) were injected in leaves. Syringe was placed against the underside of the leaf over the needle mark and pressed down gently on the plunger while directly supporting the upper side of the leaf with a finger. Liquid diffused through the leaf as it fills the mesophyllar air spaces. Firstly, the transformed *A. tumefacians* harbouring pBI121 containing *GFP* gene was injected. Therefore, transformed strains of *A. tumefacians* carrying *SrKO* and *SrUGT76G1*, were used to transform leaves individually; and for co-transformation these were mixed in equal ratio to transform stevia leaves. Samples were collected every alternate day and frozen in liquid nitrogen for further use.

## 2.6 Screening

To confirm the transformation, small pieces of infiltrated leaves were treated with acetone for four hours to remove the chlorophyll and, thus, to rule out the auto-fluorescence from the tissue. These tissues were examined for expression of *GFP* using a Leica confocal TCS SP5 microscope. DNA was extracted from 100mg each of the transiently transformed leaves and amplified for the *nptII* gene.

## 2.7 Stable transformation

To develop the stably transformed stevia plants, transiently co-transformed leaves with both pBI121-*SrUGT76G1* and pBI121-*SrKO* genes individually and together were first sterilized then inoculated on the shoot induction selection medium (SISM) containing phytohormones, NAA (0.05mg/L) and BAP (1.5mg/L) and kanamycin (10mg/L) for 3–4 weeks. Shoots were elongated on MS basal medium for 2–3 weeks. Transformants were confirmed by analyzing *nptII* gene in their genome.

## 2.8 Total RNA extraction and cDNA synthesis

Leaf tissues (100mg) were crushed under liquid nitrogen using mortar and pestle. Total RNA was extracted using Plant RNeasy Mini kit (Qiagen, Germany) following the manufacturer's protocol. Genomic DNA was digested by treating each sample with DNase I (Sigma Aldrich). The concentration and purity of RNA samples were measured by NanoDrop spectrophotometer (ND1000). The integrity was evaluated by 1.5% agarose gel electrophoresis. Each RNA sample (1 µg) was reverse transcribed in duplicate using Verso cDNA Kit (Thermo Scientific).

## 2.9 Gene normalization

Eight commonly used reference genes (RGs) (*SrβTUB*, *SrARF1*, *SrαTUB*, *SrUBQ10*, *SrSAND*, *SrGAPDH*, *SrEF-1α*, and *SrACT7*) were selected and their sequences were obtained from the

TAIR database (http://www.arabidopsis.org). Potential homologs of these RGs, were identified from the genome and transcriptome data sequences of stevia from NCBI. These homolog sequences were aligned and edited using BLAST Sequence Alignment tool. Amplicon specificity and size were verified by semi-qPCR and gel electrophoresis, respectively.

Relative expression levels were calculated using Excel-based softwares, geNORM [15], NormFinder [16], BestKeeper [17] and Reffinder [18]. To validate the results obtained from geNorm, NormFinder and BestKeeper, we applied our findings to the analysis of genes of interest in different tissues. For validation of RGs in stevia, we selected two of the best candidate RGs and the two least stable genes in the tested conditions, and compared the two RGs for their ability to provide reliable relative quantification of *Sr*UGP1 by qPCR. Trans-genes, *SrKO*, *SrUGT76G1* and other pathway related genes were analyzed by qPCR using normalized reference genes.

### 2.10 Quantitative real-time PCR assay

qPCR was carried out in optical 96-well plate with LightCycler® 480 System (Roche Diagonostics) using SYBR Green I. The qPCR was designed according to the MIQE (minimum information for publication of quantitative real-time PCR experiment) guidelines [19]. Reaction mixtures contained 10 μL SYBR Green I Mix, 2 μL diluted cDNA, double distilled water, and a final primer concentration of 0.4 μM. The following amplification conditions were applied: an initial denaturation step of 95°C for 30s; 40 cycles at 95°C for 30s; and 55°C for 30s. The final dissociation curve was obtained from 65°C to 95°C to verify primer specificity. Each assay included two technical and biological replicates. The general quality assessment of the qPCR results was based on the amplification and melting curve profiles of the samples in relation to the assay controls (non-template controls).

### 2.11 Metabolome analysis

The targeted compounds, stevioside and RebA were quantified by HPTLC and untargeted compounds were analyzed by GC-MS as per the protocol developed earlier in our lab [4, 20]. For metabolite identification and annotation, peaks were matched against customized reference spectrum databases, including the National Institute of Standards and Technology (NIST) and the Wiley Registry. Data obtained was then uploaded to the web based tool MetaboAnalyst for high throughput analysis. The data was log normalized (log2) to make it more comparable.

### 2.12 Multivariate analysis

The GC-MS data matrix after log2 normalization was used for multivariate analysis. The multivariate data matrix was analysed by MetaboAnalyst (http://www.metaboanalyst.ca/) [21]. PCA was used for an unsupervised analysis and PLS-DA for a supervised analysis. We obtained a PCA scatter plot of variables showing different metabolites to explain the separation of samples from different lines of stevia plant. A range of metabolites was selected as the variable importance in the projection (VIP) based on OPLS-DA method. Finally, Spearman's hierarchical clustering algorithm was used to group metabolites that have the same pattern of distribution.

## 3. Results

### 3.1 Transient transformation of stevia leaf by recombinant vectors

*Agrobacterium tumefaciens* strain EHA105 harbouring binary construct pBI121-*SrKO* and pBI121-*SrUGT76G1* were used to transiently transform the leaves of stevia pBI121 harbouring

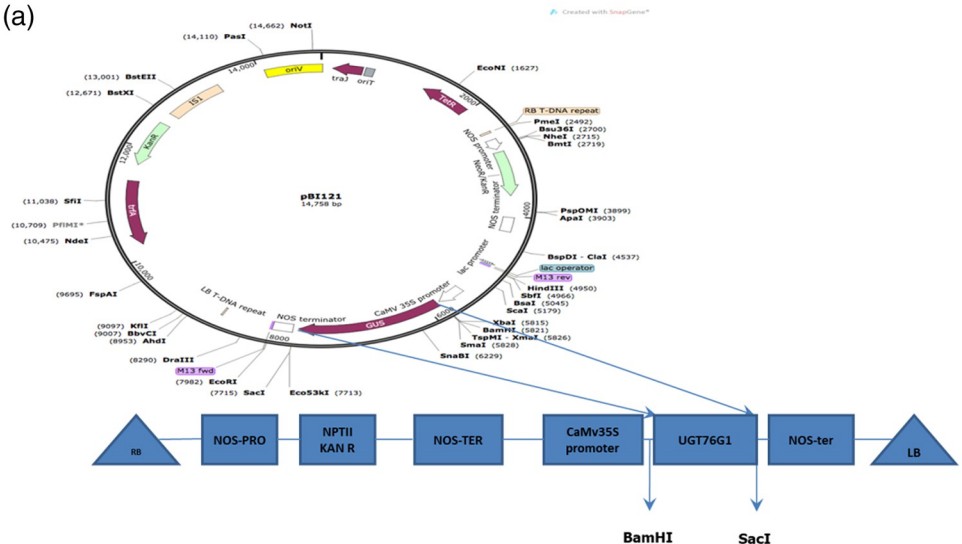

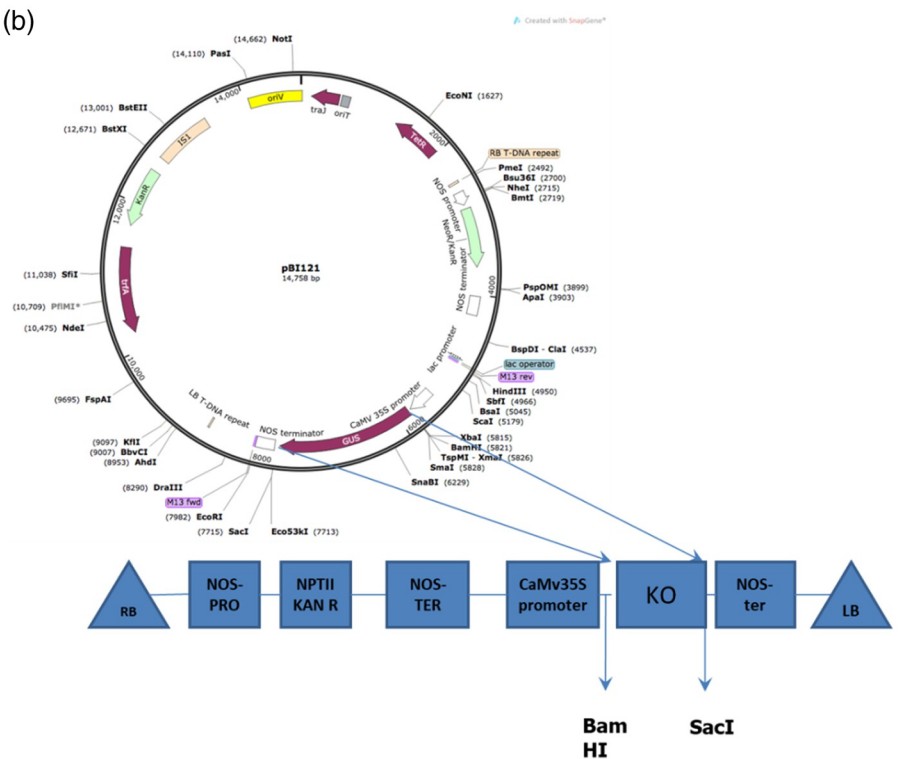

**Fig 1. Binary vectors containing A: *UGT76G1* and B: *KO* gene inserted in place of *gus* gene.**

*GFP* as reporter gene was used as control (Fig 1a and 1b). The transformed leaves were confirmed by visualizing the leaves under Leica TCS SP5 confocal microscope equipped with long lasting solid state lasers (Leica Microsystems, Germany). Expression of *GFP* was visible after 24h of transformation, which increased with time until reached a plateau phase at fourth day (Fig 2a and 2b) there, was no florescence in control (Fig 2c). The *SrKO* and *SrUGT76G1* cDNA were cloned separately in pBI121 plasmids. These plasmids were then mobilized in *A.*

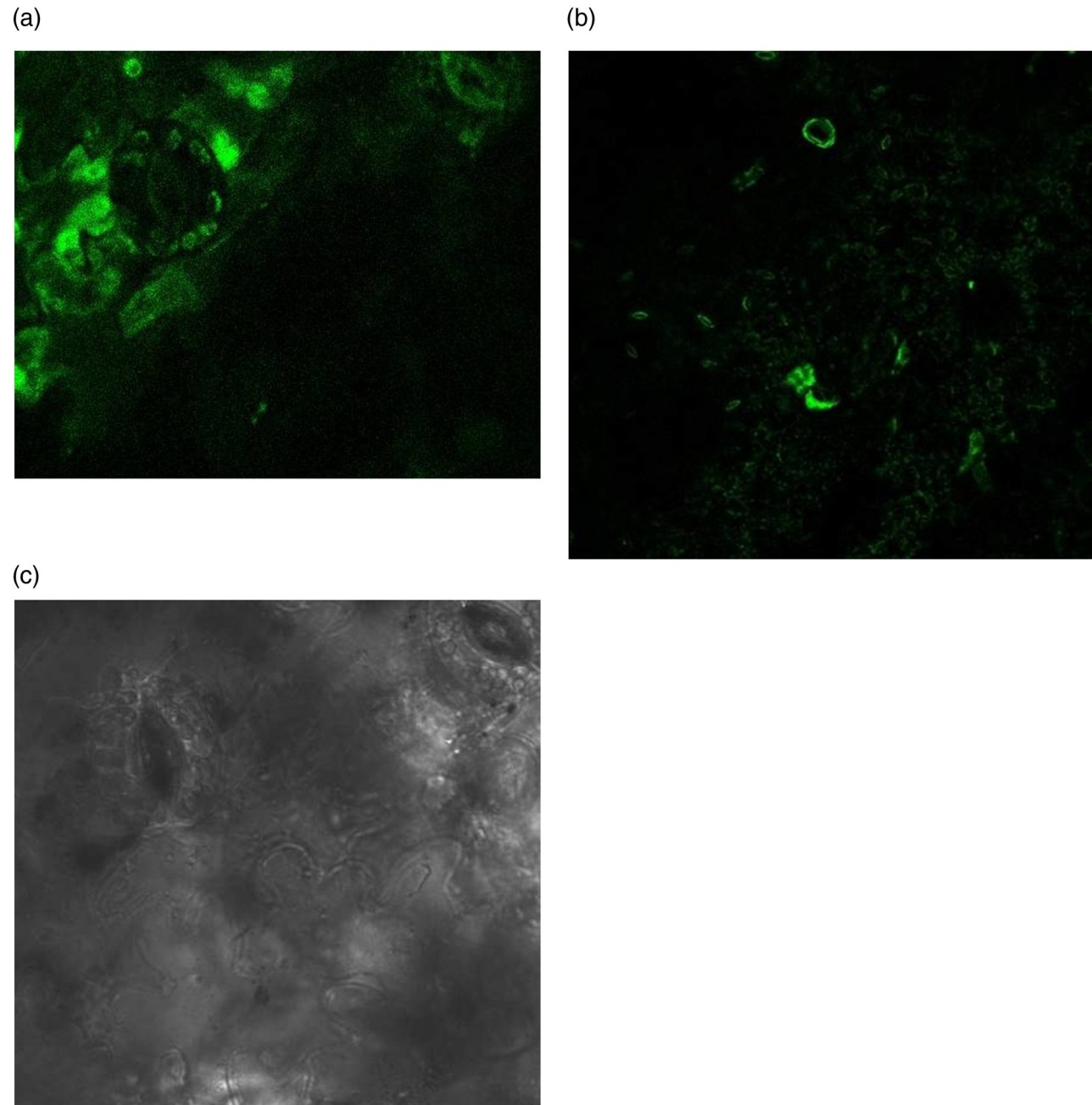

(a)

(b)

(c)

**Fig 2. *GFP* visualization through confocal microscope A, B: Leaves transformed with *GFP*, C: Control leaves without *GFP*.**

*tumefacians* strain EHA105. The transformed *A. tumefacians* EHA105 were maintained on culture medium separately for transformation of stevia leaves with individual *SrKO* and *SrUGT76G1* and mixed in 1:1 ratio to co-transform stevia leaves with both *SrKO* and *SrUGT76G1* cDNA. Transient transformation of the leaves was confirmed by the amplifying genomes of stevia using CAMV-35S/*SrKO* and CAMV-35S/*SrUGT76G1* specific primers. Co-transformation of stevia leaves with *SrKO* and *SrUGT76G1* was also confirmed by presence of both the amplicons of 150bp and 200bp, respectively in the genome of stevia (Fig 3). HPTLC

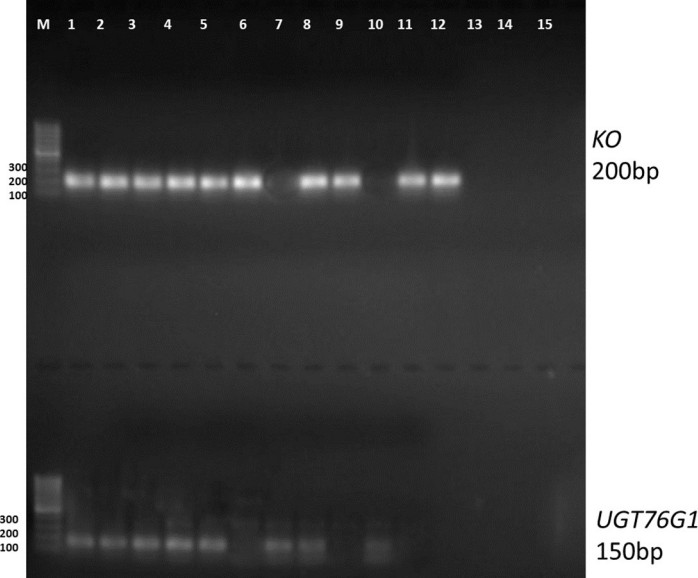

**Fig 3. Transformation confirmation by visualizing the desired amplicons on agrose gel electrophoresis.**

was used to investigate the changes in stevioside and rebaudioside-A contents, the two industrially important glycosides in transiently transformed stevia leaves. When *KO* gene was expressed alone in transiently transformed leaves, stevioside on an average increased by 23.35%, rebaudioside-A increased by 11.22% and rebaudioside-A to stevioside ratio was found to be 0.43 (Fig 4a), almost same as that of the control (0.41). Overexpression of *UGT76G1* gene resulted in decrease in concentration of stevioside by 10.43%, increase in rebaudioside-A by 15% and change in the ratio of rebaudioside-A to stevioside from 0.43 to 0.52 (Fig 4b). When both the genes (*KO* and *UGT76G1*) were co-expressed, it led to the decrease in stevioside content by 8.5% and increase in rebaudioside-A by 69.9%. Rebaudioside-A to stevioside ratio was higher in transiently co-transformed leaves with both *KO-UGT76G1* genes (1.23), followed by the leaves transformed with either *UGT76G1* (0.52) or *KO* gene (0.43), respectively when compared with control (0.41) (Fig 4c).

## 3.2 Stable plant lines from transiently transformed leaves

RebA to stevioside ratio was greater in co-transformed leaves, thus they were used for stable transformation. Transiently co-transformed leaf discs (2cm$^2$) of stevia with both *SrKO* and *SrUGT76G1* transcripts were surface sterilized and then cultured for 3–4 weeks on SISM (shoot induction selection medium) consisting of MS basal medium containing 1.5mg/L BAP, 0.05mg/L NAA and 10mg/L kanamycin for selection. Untransformed leaf discs died due to the lack of kanamycin resistant gene (*nptII*) whereas, positively transformed ones regenerate on the selection medium. These regenerated transforments were then further elongated on MS basal medium for 3–4 weeks. The roots were induced after 7–10 days of transfer (Fig 5).

## 3.3 Molecular analysis of transformed and non-transformed stevia plants

Genomic DNA extracted from all putative transgenic lines were subjected to PCR analysis to detect the presence of CAMV 35S/*SrUGT76G1* (150bp) and CAMV 35S/*SrKO* (200bp) specific amplicons. Transgenic lines positive for both the genes were further confirmed by the presence

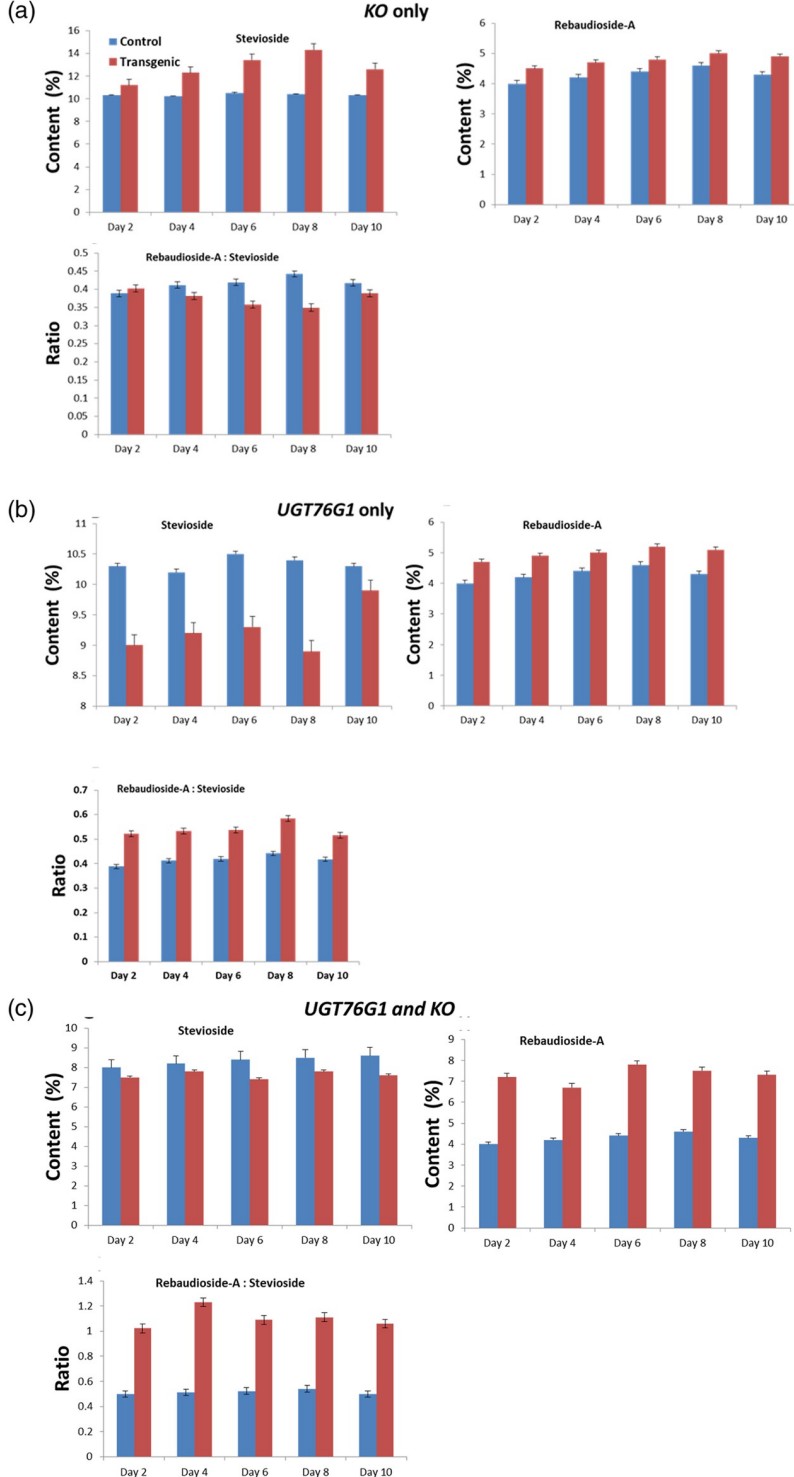

**Fig 4. Concentration of stevioside and rebaudioside-A after overexpression of genes *KO* and *UGT76G1*.** A: only *KO* gene was overexpressed, B: only UGT76G1gene was overexpressed, C: UGT76G1 and KO genes were co-expressed.

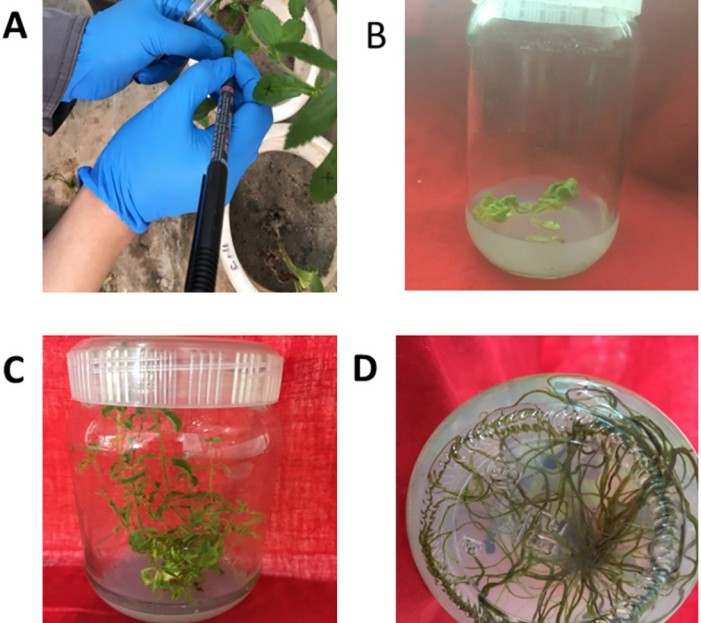

**Fig 5. In vitro culture of Stevia rebaudiana A: Transiently transfored leaves B: Transiently transformed leaves cultured on SISM (shoot induction selection medium) (MS basal medium + 1.5mg/L BAP + 0.05mg/L NAA + 10mg/L kanamycin) C: Multiple shoots on SISM And D: Roots on MS basal medium.**

of *nptII* gene (300bp). Out of 10 putative transgenic lines, 7 transgenic lines were positive for both the genes. No amplification was, however, detected in the nontransgenic plant of stevia (Fig 6).

## 3.4 Morphological analysis

The shoot length of both the transgenic and non-transgenic stevia plants were measured with a scale in centimeter and the number of the leaves per plant were counted after fourth week of culture on the rooting medium. The shoot length was 6.5 ± 0.54 on an average which is similar to the control plants on the 4th week of culture on rooting medium. Similarly, numbers of leaves per plant were 23.2 ± 0.3 in plants of the transgenic lines over-expressing both *SrKO* and *SrUGT76G1* comparable with the non-transgenic plants. Plants with morphological

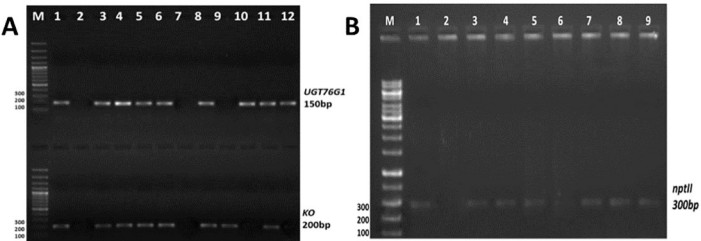

**Fig 6.** A: Confirmation of putative transgenic lines by amplifying the *UGT76G1* and *KO* specific primers. Out of 10 transgenic lines, only 7 were positive for both the genes, B: transgenic lines which were positive for both the genes were further confirmed by the presence of *nptII* gene. M = 100b ladder (Fermantas) Lan 1 = positive control, Lane 2 = non-transformed control, Lines for *nptII* were selected for further analysis.

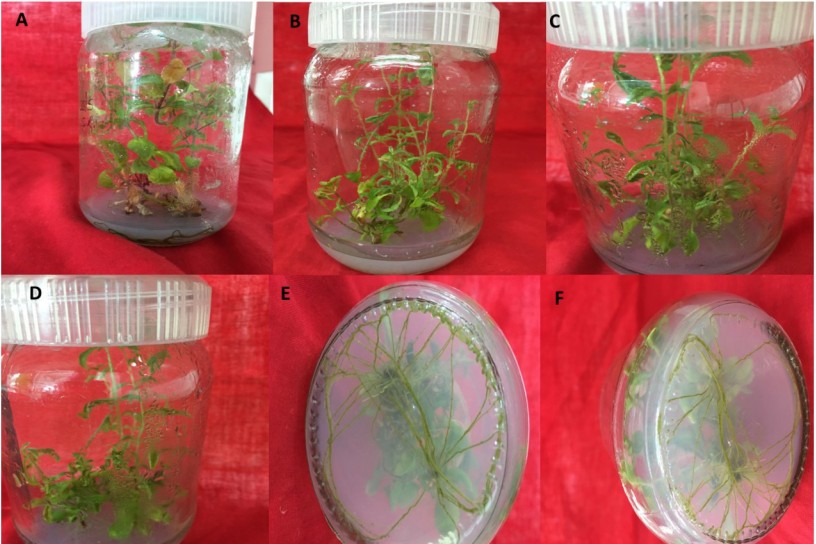

**Fig 7. *In vitro* developed transgenic plants.** A: Non-transgenic plant, B-D: Transgenic plants that are morphologically similar to non-transgenic plants. E: Roots of non-transgenic plant and F: Roots of transgenic plant.

malformations were discarded. Plants with no significant difference in these parameters were further analyzed (Fig 7, Table 2).

### 3.5 Reference gene normalization

**3.5.1 Quality control and selection of candidate reference genes.** *Agrobacterium* and transient genes are generally perceived as foreign genes by the plant cells, thereby inducing defense response. *Agrobacterium* uses a complex set of its own virulence functions to genetically transform the host cells, and actively interfere with some of the basic cellular processes to escape host defense. Monitoring changes in transcriptome of *Arabidopsis* from 4-48h period after *Agrobacterium* infection, Ditt *et al*, in 2006 found significant changes in various defense response genes including downregulation of expression of actin and ubiquitin (commonly used as reference genes) [22]. Raw gene expression data obtained from transgenic stevia plants suggest the profound effect of transgenes on various reference genes. Collectively, validating the earlier reports that no single reference gene is consistent across all experiments [23, 24].

**Table 2. Shoot length (cm) and number of leaves in non-transgenic and transgenic plants of *Stevia rebaudiana* on fourth week of culture.**

| Genotype | Shoot length(cm) | No. of Leaves |
|---|---|---|
| Control | 6.05 ±0.12 | 23.33 ±0.72 |
| KU1 | 6.15 ±0.44 | 22.42 ±0.82 |
| KU2 | 6.50 ±0.65 | 24.23 ±0.29 |
| KU3 | 5.43 ±0.54 | 20.22 ±0.54 |
| KU4 | 6.71 ±0.23 | 22.51 ±0.67 |
| KU5 | 5.34 ±0.52 | 23.44 ±0.97 |
| KU6 | 6.8 ±0.14 | 22.81 ±0.87 |
| KU7 | 6.77 ±0.56 | 21.78 ±0.65 |

Each value is the mean ± standard deviation (n = 3).

**Table 3. Primer sequences of eight reference genes along with their accession numbers and *A. thaliana* orthologs.**

| GENE | Accession | *A. thaliana* ortholog | Annotation | Primer sequence | | GC content/ % | Melting Temp. (°C) |
|---|---|---|---|---|---|---|---|
| *Sr*GAPDH | BG521434 | AT1G13440 | glyceraldehyde-3-phosphate dehydrogenase | F: | GGAGCTGAGTATGTTGTGGAA | 47.6 | 57 |
| | | | | R: | GCGGAGATGATGACCTTCTTAG | 50 | 60 |
| *Sr*UBQ10 | BG522957 | AT4G05320 | polyubiquitin 10 | F: | TATCCCACCAGACCAACAAAG | 50 | 62 |
| | | | | R: | GGACAAGATGGAGAGTGGATTC | 50 | 62 |
| *Sr*ACT7 | AF548026 | AT5G09810 | Actin 7 | F: | CCCAAGGCGAACAGAGAAAAG | 52.4 | 65 |
| | | | | R: | TGTACGACCACTGGCATAAAG | 47.6 | 59 |
| *Sr*EF1A | AY157315 | AT1G07920 | Elongation factor | F: | GCTCTTCTTGCTTTCACTCTTG | 45.4 | 59 |
| | | | | R: | GATTTCTTCATACCTCGCCTTTG | 43.4 | 58 |
| *Sr*βTUB | BG526269 | AT2G29550 | Beta tubulin | F: | GCTCTTCTTGCTTTCACTCTTG | 45.4 | 59 |
| | | | | R: | CGCCTCGTTATCAAGAACCATA | 45.4 | 59 |
| *Sr*SAND | BG521548 | AT2G28390 | SAND family protein | F: | CCGTGTCTTCCTCTTGCTTATG | 50 | 62 |
| | | | | R: | ACCACCTTATCTTTGGCACATC | 45.4 | 58 |
| *Sr*αTUB | BG524658 | AT5G19770 | Tubulin alpha-3 | F: | GGTGATGTTGTCCCGAAAGA | 50 | 61 |
| | | | | R: | GACTGTTGGTGGCTGATAGTT | 47.6 | 55 |
| *Sr*ARF1 | BG523226 | AT1G23490 | ADP-ribosylation factor | F: | CCAGAACACACAGGGTCTTATC | 50 | 62 |
| | | | | R: | CACAAGGAGAACTGCATCTCT | 47.6 | 59 |

Henceforth, before transgene expression quantification, the reference genes were normalized to increase experimental consistency, integrity and transparency. From the available dbEST database of stevia, a series of candidate reference gene sequences were retrieved. Eight potential candidate reference genes namely, *SrACT7, SrGAPDH, SrβTUB, SrEF-1α, SrSAND, SrARF1, SrαTUB* and *SrUBQ10* were identified for gene expression studies using qPCR in stevia (Table 3). Specific primers were designed and confirmed on the basis of the amplification specificity and efficiency results of the candidate reference genes. The amplified PCR products using specific primers of the genes yielded amplicons of the expected sizes that were confirmed by resolving the amplified products on 1.2% agarose gel electrophoresis. The presence of single band and single-peak melting curves obtained in each case ruled out the presence of primer dimers.

### 3.5.2 Expression profiles of candidate reference genes

RNAs isolated from all plant samples were reverse transcribed to cDNAs for further use in qPCR detection. The Cq values of 8 reference genes were determined in all transiently transformed leaf samples by *A. tumefacians*. Cq values showed differential transcript levels in all the tissue samples, including transformed leaf examined. The lower Cq value corresponds to higher transcript abundance and vice versa. The mean Cq values of 8 potential reference genes ranged from 12.21 to 30.61. A box and whiskers plot was used to determine the raw data distribution (Fig 8). In the transformed leaf, *SrUBQ10* has a minimum average Cq value of 15.62, while *SrSAND* has a maximum average Cq value of 22.83. Further, CV of the Cq values was also calculated to evaluate the expression levels of candidate reference genes under all experimental sets. In four statistical algorithms (geNorm, NormFinder, BestKeeper and RefFinder) candidate reference genes were ranked differently, based on their gene expression stability in transgenic stevia plants. There was subtle difference in the overall trend of most and least stable reference genes, which might be ascribed to the difference in algorithms.

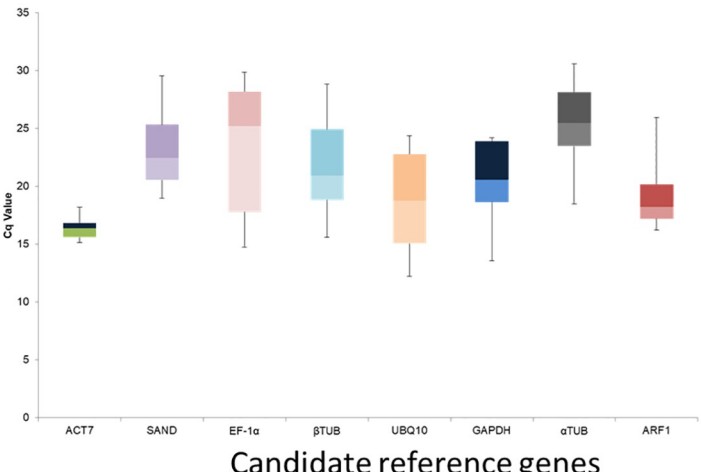

**Fig 8. Cq values of eight candidate reference genes in all the samples and *Agrobacterium* mediated transiently transformed leaves were described using box and wiskers plot.** The outer box is determined from 25th to 75th percentile, and the inner box represents the mean value. The whisker caps represent the minimum and maximum values.

**3.5.3 geNorm analysis.** geNorm is the strong algorithm for large and small sample size. It analyzes the best reference gene by pairwise correlation among different reference genes. geNorm ranks the candidate reference genes according to their expression stability. This algorithm provides with the best reference genes by repeated process of stepwise exclusion of the highest pairwise variation [17]. Average pairwise variations (Vn/n + 1) are calculated between the normalization factors, NFn and NFn + 1. If pairwise variation is > 0.15, it is not necessary to use ≥ n + 1 as an internal control for particular experimental conditions. geNorm, a statistical algorithm was used to analyze the expression stabilities of the eight candidate reference genes. geNorm calculates gene expression stability (M) as the average pair-wise variation between all tested genes in a given set of samples. According to the geNorm analysis, the cut-off range of stability value (M) is < 1.5, so lower the M value most stable is the reference gene in terms of gene expression and vice versa. When all the samples were analysed *Sr*SAND and *Sr*GAPDH with M value of 0.852 were found more stable and pattern followed was, *SrARF1 < SrUBQ10 < SrEF-1α < SrACT7 < SrαTUB < SrβTUB < SrSAND < SrGAPDH* (Fig 9). Many researchers use only one reference gene for normalization. Over the years of research it is evident that use of single reference gene is not suitable for gene expression analysis. Even in our case, use of two most stable reference genes would be mandatory for normalization purpose.

**3.5.4 NormFinder analysis.** NormFinder is an excel based mathematical tool that ranks the best reference genes according to their minimal combined inter and intra group expression variations. Lower average expression stability values indicated genes with more stable expression. This method calculates inter and intra-group variance. Inter-group expression variation allows the selection of best suited reference gene, while intra-group variance estimates the minimum number of genes to be included in a particular experiment. The results of the NormFinder analysis are shown in Table 4. In transiently transformed leafs, *SrGAPDH* with stability factor of 0.295 was most stably expressed. *SrSAND* and *SrGAPDH* were therefore recommended in combination for normalization of reference genes in these tissues. The expression stability ranking of NormFinder analysis of the eight reference genes of interest were relatively consistent with the data array of geNorm, strengthening the validity of these methods.

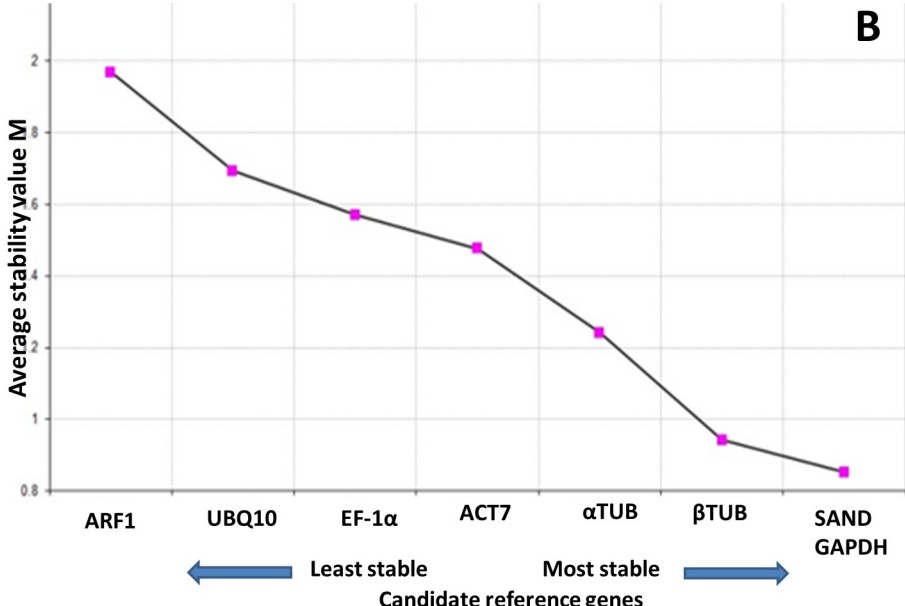

**Fig 9. Gene expression stability values (M).** Ranking of the gene expression stability performed in *A. tumefacians* mediated transiently transformed leaves.

**3.5.5 BestKeeper analysis.** BestKeeper software, an excel based tool estimates inter gene relations of possible reference gene pairs by performing numerous pairwise correlation analysis using raw Cq values of each gene. BestKeeper evaluates gene expression stability for individual reference genes based on standard deviation (SD), coefficient of correlation and percentage covariance. Reference genes with SD values greater than 1 are considered as inconsistent and should be excluded. For the transiently transformed leaves, the candidate reference genes, *SrSAND*, *SrEF-1α*, *SrβTUB*, *SrUBQ10*, *SrGAPDH*, *SrαTUB* and *SrARF1* showed SD values >1.0. *SrACT7* with SD value of 0.49 thus, was identified as the most reliable reference combination for normalization.

**3.5.6 RefFinder analysis.** RefFinder (http://www.leonxie.com/referencegene.php) is user friendly web-based software that compares data generated from different softwares (geNorm, NormFinder, ΔCt and BestKeeper) and gives comprehensive ranking to confirm the stability via different programs. It assigns an appropriate weight to an individual gene from each program and calculates the geometric mean of their weights for the overall final ranking. The Cq

**Table 4. Candidate reference genes and best combination of two genes listed according to their expression stability calculated by NormFinder in transiently transformed leaves.**

| Genes | Expression stability value |
|---|---|
| *Sr*GAPDH | 0.295 |
| *Sr*βTUB | 0.479 |
| *Sr*SAND | 0.610 |
| *Sr*ACT7 | 0.990 |
| *Sr*αTUB | 1.021 |
| *Sr*EF-1α | 1.067 |
| *Sr*UBQ10 | 1.190 |
| *Sr*ARF1 | 1.756 |

values were fed into the program directly. Data generated by geNorm, NormFinder, Best-Keeper across all sets were then compared by RefFinder. *Sr*GAPDH with geomean ranking values of 1.31 was stably expressed in transiently transformed leaves.

To validate the reference genes the best (*Sr*GAPDH and *SrSAND*) and the worst (*Sr*ARF1) reference genes were used to estimate the expression of *Sr*UGP1. When *Sr*GAPDH and *SrSAND* genes were used for normalization, the expression of the gene *Sr*UGP1 showed higher expression in older leaves than young ones as found in earlier studies [4], whereas there was not much difference in expression when normalized with worst reference gene. Thus, this validates our reference genes.

**3.5.7 Gene expression analysis.** Transformed plants selected were further analyzed for the change in the metabolites contents and the gene expression. Key genes, namely *SrKO*, *Sr*KS, *SrUGT76G1*, *Sr*UGT74G1, *Sr*UGT85C2, *Sr*DXR, *Sr*KAH, *Sr*UGP1, *Sr*LMS and *Sr*NMD were selected for the gene expression studies using qPCR in transformed and control (non-transformed) stevia plants. The amplified PCR products of the expected sizes using gene-specific primers were confirmed by resolving on 1.2% agarose gel. The Cq values of all the genes determined in all the tissue samples by qPCR. Cq values obtained were suggestive of the differential transcript levels in all the tissue samples. The lower Cq value corresponds to higher transcript abundance and vice versa. The mean Cq values of all the genes ranged from 12.21 to 30.61. RNA transcript levels varied considerably from one transgenic line to the other. Compared with nontransgenic (control) plants, the expression of *Sr*DXR, *Sr*KS, *SrKO*, *Sr*KAH, *Sr*UTGT76G1, *Sr*UGT85C2 and *Sr*UGT74G1 genes increased in transgenic plants. The expression of *Sr*NMD and *Sr*LMS genes, however, decreased in transgenic plants (Table 5, Fig 10).

**3.5.8 Metabolome profiling.** To investigate the impact of *SrKO* and *SrUGT76G1* overexpression at metabolic level, targeted profiling of stevioside and RebA was done by HPTLC and untargeted metabolite profiling was done by GC-MS. Leaves of transgenic and non-transgenic plants were harvested for metabolite analysis. Chromatographic peaks of stevia leaves extracted in methanol were identified using the NIST and Wiley databases. Preliminary GC-MS metabolites acquired from all the samples were >350 resolved peaks that are shown as an overlay in Fig 11. Nearly 55% of these peaks could be identified as distinct metabolites with known chemical structure. These identified metabolites represent numerous metabolic pathways, including biosynthesis of steviol glycoside, phytosterols, (+)-isomenthol and fatty acids biosynthetic pathways.

**3.5.9 Analysis of metabolites.** To evaluate the metabolites, data were log normalized to make them more comparable. Initial analysis of metabolites obtained from HPTLC and GC-MS showed a change in metabolite concentrations in different transgenic lines. Using *t*-test metabolites with significant change (P <0.05) were screened in transgenic lines. The level of two compounds of our interest, steviol and RebA showed significant increase in transgenic lines when compared with control. Stevioside content as per our expectation decreased in all the transgenic lines (Fig 12). The RebA to stevioside ratio increased in all the transgenic lines but in KU2 line there was a significant increase 2.83 compared to 0.74 in control. Next PCA was performed for transgenic and control plants for all the metabolites. The PCA score plot exhibited a clear separation between control vs transgenic plants. Many untargeted compounds were significantly different between transgenic and control plants (Fig 13). PCA 1 had 65.3% variance PCA2 had 11.1% variance. Along with targeted compounds, steviol, stevioside and RebA, other untargeted compounds showed significant variance between transgenic and non-transgenic plants.

Partial least squares-discriminant analysis (PLS-DA), orthogonal projection to latent structures discriminate analysis (OPLS-DA) and Heatmap analysis was done. Score plots from the supervised OPLS-DA showed obvious separation of metabolites between control and

**Table 5. Sample spreadsheet of data analysis using the 2-$^{\Delta\Delta ct}$ method [25].**

| Sample | *UGT76G1* | *UGT74G1* | *UGT85C2* | *UGP1* | *KAH* | *KS* | *KO* | *DXR* | *NMD* | *LMS* |
|---|---|---|---|---|---|---|---|---|---|---|
| KU1 | 0.403 | 0.353 | 0.360 | 0.570 | 0.683 | 0.413 | 0.480 | 2.713 | 1.867 | 3.653 |
| | ±0.07 | ±0.09 | ±0.04 | ±0.05 | ±0.03 | ±0.04 | ±0.03 | ±0.19 | ±0.12 | ±0.29 |
| KU2 | 0.573 | 0.717 | 0.660 | 1.547 | 0.880 | 0.837 | 0.743 | 4.483 | 1.213 | 2.823 |
| | ±0.05 | ±0.07 | ±0.05 | ±0.12 | ±0.10 | ±0.06 | ±0.06 | ±0.44 | ±0.15 | ±0.16 |
| KU3 | 3.187 | 3.157 | 3.443 | 5.503 | 4.543 | 2.077 | 1.633 | 6.257 | 0.327 | 0.473 |
| | ±0.15 | ±0.13 | ±0.19 | ±0.06 | ±0.07 | ±0.22 | ±0.13 | ±0.06 | ±0.02 | ±0.04 |
| KU4 | 2.510 | 3.587 | 3.303 | 6.390 | 5.210 | 2.363 | 2.033 | 7.310 | 0.513 | 0.530 |
| | ±0.27 | ±0.16 | ±0.11 | ±0.10 | ±0.11 | ±0.05 | ±0.14 | ±0.11 | ±0.02 | ±0.02 |
| KU5 | 1.633 | 2.943 | 2.167 | 4.400 | 3.913 | 1.730 | 1.347 | 6.663 | 0.657 | 0.903 |
| | ±0.13 | ±0.17 | ±0.48 | ±0.48 | ±0.33 | ±0.13 | ±0.09 | ±0.28 | ±0.05 | ±0.07 |
| KU6 | 1.227 | 2.477 | 1.157 | 3.663 | 2.117 | 1.493 | 1.247 | 6.767 | 0.793 | 1.390 |
| | ±0.10 | ±0.06 | ±0.09 | ±0.07 | ±0.05 | ±0.05 | ±0.10 | ±0.11 | ±0.08 | ±0.17 |
| KU7 | 0.737 | 1.803 | 0.863 | 3.023 | 1.447 | 1.260 | 0.887 | 5.587 | 1.083 | 2.420 |
| | ±0.10 | ±0.27 | ±0.10 | ±0.11 | ±0.16 | ±0.16 | ±0.08 | ±0.06 | ±0.11 | ±0.10 |

The fold change in expression of the target gene relative to the internal control gene (*SrGAPDH* and *SrUBI*) in seven transgenic lines was studied. The samples were analyzed using real-time quantitative PCR.

transgenic lines (Fig 14). The heatmap, commonly used for unsupervised clustering, was constructed based on OPLS-DA analysis. The heatmap showed a differential distribution of metabolites in transformed plants Steviol and its derivatives exhibited significant increase, while the accumulation pattern of each metabolite was different in different lines. Compared to the control plants, 13 metabolites were up-regulated 20 downregulated significantly in all the transgenic lines (Fig 15).

### 3.5.10 Metabolic changes and correlation of metabolite levels

Correlation analysis is an important tool to explore metabolic pathways and networks. Spearman's correlation for each metabolite in different lines demarcated the changes in transgenic stevia. Correlation analysis of the 61 differential metabolites, marked on the hierarchical clustering plot, was performed to understand the potential relationships between metabolites. RebA showed a positive correlation with steviol (0.53), Stevioside (r = 0.65) (Fig 16).

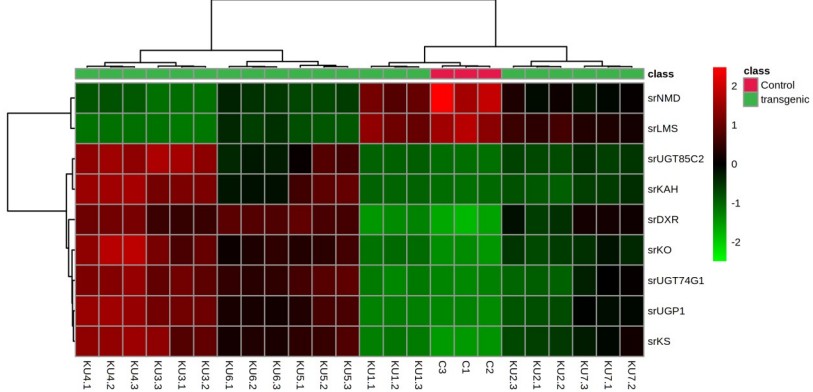

**Fig 10. Differential expression of genes in control and transgenic plants.** Color scale represents gene expression with red being highest and green lowest.

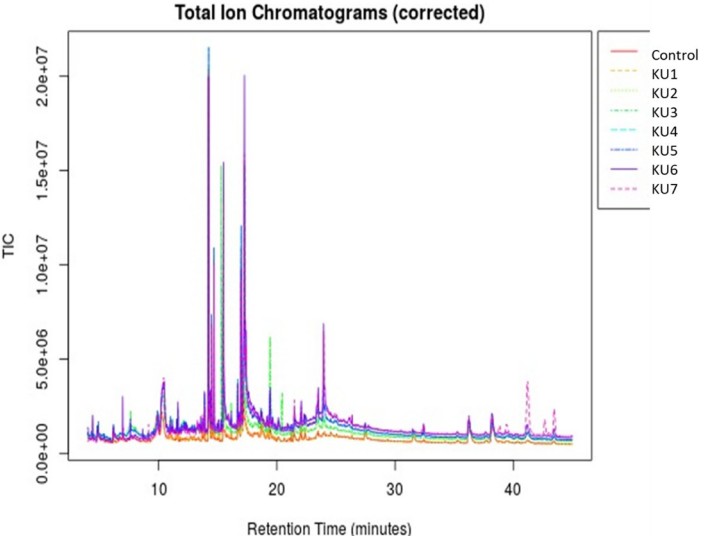

**Fig 11. Overlay of peaks of different metabolites of transgenic and control stevia lines obtained from GC-MS.**

## 4. Discussion

Stevia is the only plant which produces steviol glycosides. Steviol glycosides have very unequalled properties unique to them. They are 200–300 times sweeter than sugar. Additionally, recent studies have proved the insulinotropic nature of steviol, stevioside and RebA [26]. Among all the steviol glycosides RebA is the sweetest of them all. But RebA is less abundant than stevioside which is bitter. Thus we overexpressed two main pathway genes *SrKO* and *SrUGT76G1* to increase the ratio of RebA to stevioside.

Genes *SrKO* and *SrUGT76G1* were transiently transformed individually and co-transformed in stevia leaves. To make sense of gene expression, we normalized reference genes for our experiment according to the MIQE guidelines. We used three excel based software geNORM [17], NormFinder [27] and BestKeeper, operating on different algorithms and Reffinder [18] (a web-based tool) and ΔCt method to analyze the stably expressed reference genes in stevia. From the available dbEST database of stevia, a series of candidate reference gene sequences were retrieved. After analyzing the primer amplification, eight genes were selected to determine their stability.

Final ranking of the reference genes was slightly different in each statistical algorithm, but there was significant agreement between the genes with the most and least stable expression. Combination of *SrGAPDH* and *SrSAND* are most suitable for in transformed leaves of stevia. Using these reference genes, we found that expression of transgenes showed a significant increase in expression.

Overexpression of *SrKO* transgene showed an increase in steviol, stevioside and RebA but there was no change in RebA to stevioside ratio. The previous studies also suggest that the gene expression of *SrUGT76G1* is highly correlated with the RebA content in stevia plant leaves [28]. In case of overexpression of *SrUGT76G1*, RebA to stevioside showed a significant increase. However, when both the genes *SrKO* and *SrUGT76G1* were co-expressed, it led to the decrease in stevioside content by 8.5% and the increase in rebaudioside-A by 69.9%. Rebaudioside-A to stevioside ratio was higher in transiently co-transformed leaves with *SrKO* and *SrUGT76G1* genes both (1.23), followed by the leaves transformed with either *SrUGT76G1* (0.52) or *SrKO* gene (0.43), when compared with control (0.41). Varieties of stevia with high

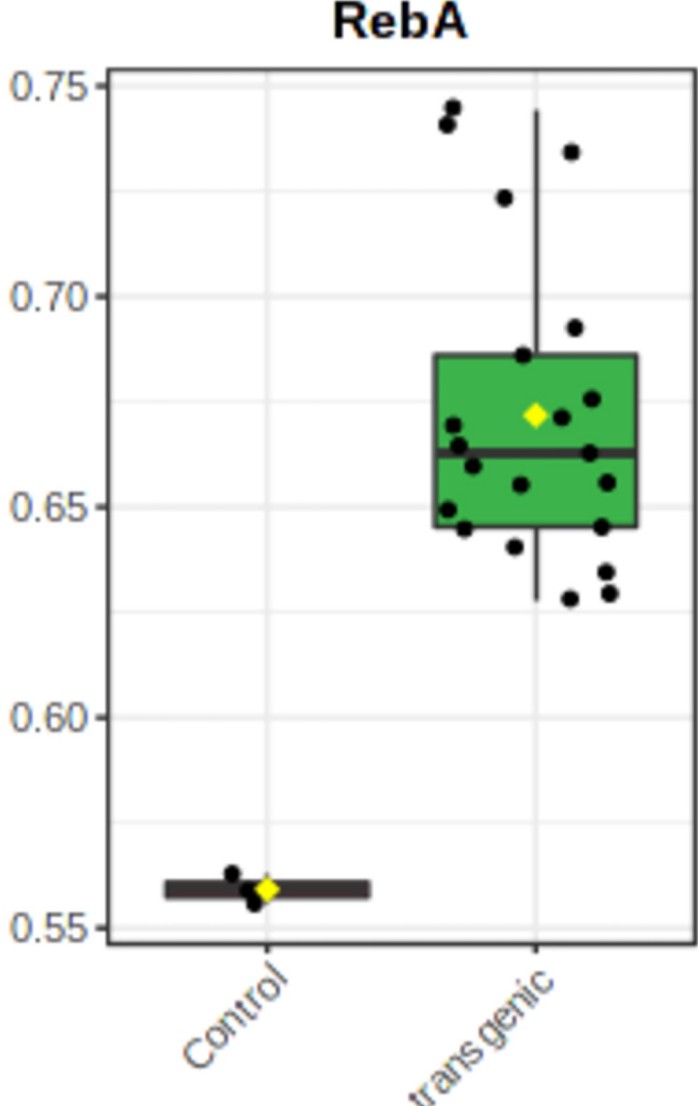

**Fig 12. Concentration of RebA and Steviol showed increased significantly, whereas stevioside showed significant decrease in all the transgenic lines.**

ratio of RebA to stevioside in their leaves had highly active *SrUGT76G1* enzyme [29]. Transiently transformed leaves from each sample were equally pooled together to establish the results. In stevia, *SrKO* is expressed in both young and mature leaf tissues. Gene expression of *SrKO* is highly correlated with the metabolite concentrations of steviol glycosides [4].

Similarly, there is a significant correlation between the higher SGs accumulation and the transcription of *SrUGT76G1* [12].

Leaves which were transiently co-transformed with *SrKO* and *SrUGT76G1* were used to generate stable plants. Leaves were surface sterilized and placed on the MS basal media for 2 days to stabilize and acclimatize under *in vitro* conditions. After 2 days, explants were transferred to the optimized regeneration medium with appropriate antibiotics (Kn 10mg/L). After 3–4 weeks, the putative transgenic shoots were transferred to elongation medium. These shoots showed significant growth on MS basal medium and rooting was also initiated on same

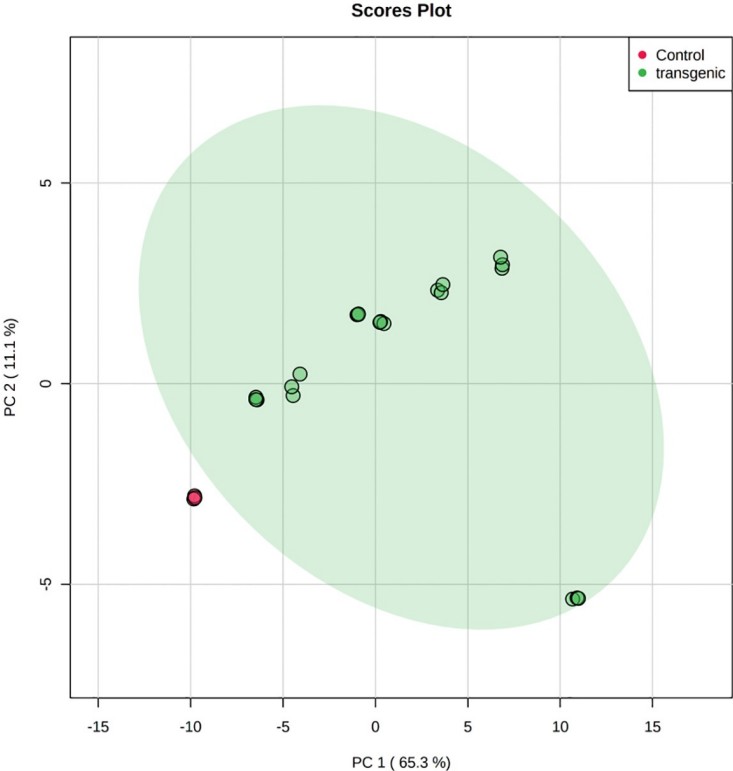

**Fig 13. PCA scatter plots of variables showing different metabolites to explain the separation of samples from different lines.** PCA score plot of samples from control and transgenic lines. Each with triplicate samples taken for extraction at each time point, were used in the analysis, for a total of 24 data points.

medium after 3–4 weeks. Putative transgenic plants were further confirmed by the presence of *nptII* gene of 300kb on 1.0% agarose gel. Out of these, 70% of the plants were found transgenic. The co-transformation of *SrKO* and *SrUGT76G1* was confirmed by the presence of CaMV-35S/*UGT76G1* (150bp) and CaMV-35S/*KO* (200bp) specific gene bands. The percentages of transgenic plants obtained using transiently transformed method were significantly higher than the conventional transformation protocols as reported earlier [30].

In confirmed transgenic lines expression of transgenes and seven other important genes were analyzed by qPCR. Genes *SrDXR*, *SrKS*, *SrKO*, *SrKAH* and *SrUGTs*, were chosen as they are directly involved in steviol glycosides biosynthesis pathway [31]. *SrLMS* and *SrNMD* genes participate in monoterpenoid biosynthesis [32], whereas SrUGP1 is needed in every glycosylation steps in steviol biosynthetic pathway [33]. Expression of *SrKO* and *SrUGT76G1* increased in all the transgenic lines. RNA transcript levels varied considerably from plant to plant, with the expression of genes related to steviol glycoside pathway (*SrKO*, *SrKS*, *SrUGT76G1*, *SrUGT74G1*, *SrUGT85C2*, *SrDXR*, *SrKAH* and *SrUDP1*). The expression of these genes showed a considerable increase in all the transgenic lines, whereas *SrLMS* and *SrNMD* somehow showed a slight decrease in expression when compared with control (non-transgenic) plant.

Higher transgene expression led to the increased concentration of RebA in all the seven transgenic lines (KU1-KU7) than control plant and RebA to stevioside ratio also increased significantly. Steviol, stevioside and RebA showed a differential concentration in all the seven lines, but the pattern was the same in all of them and the ratio of stevioside increased dramatically. In transgenic line 2 (KU2), RebA showed a steep increase in concentration 52% the

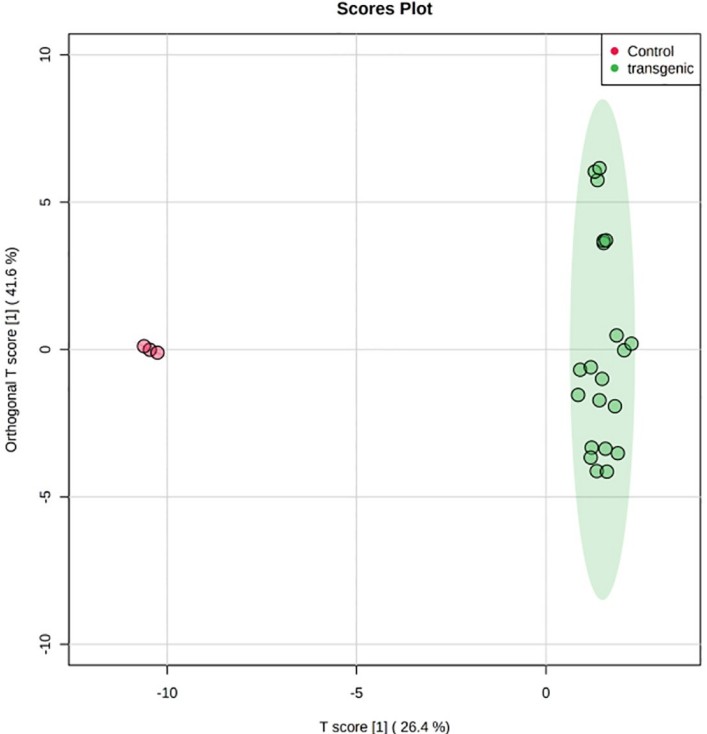

**Fig 14. OPLS-DA plot of control and transgenic lines on the leaf metabolic profiles.**

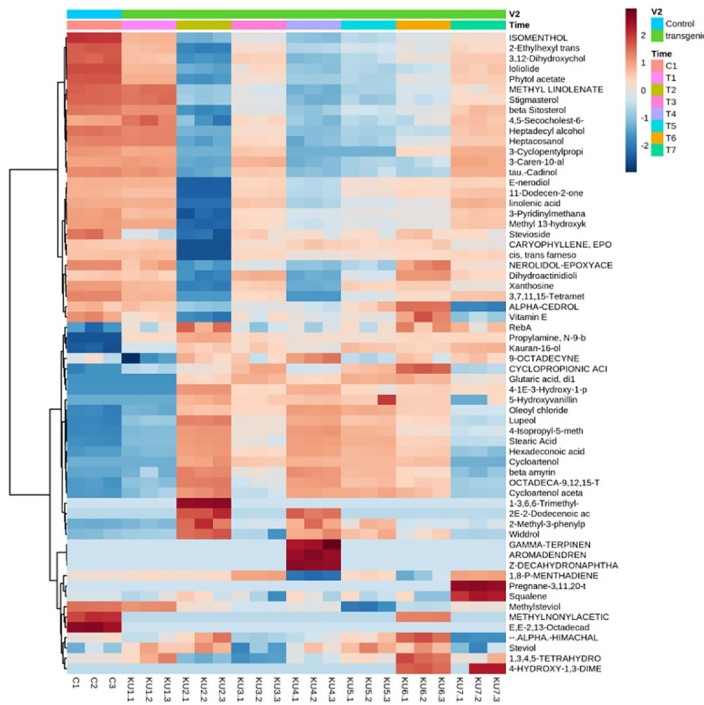

**Fig 15. Correlation analysis of the 61 differential metabolites, marked on the hierarchical clustering plot to understand the potential relationships among metabolites.**

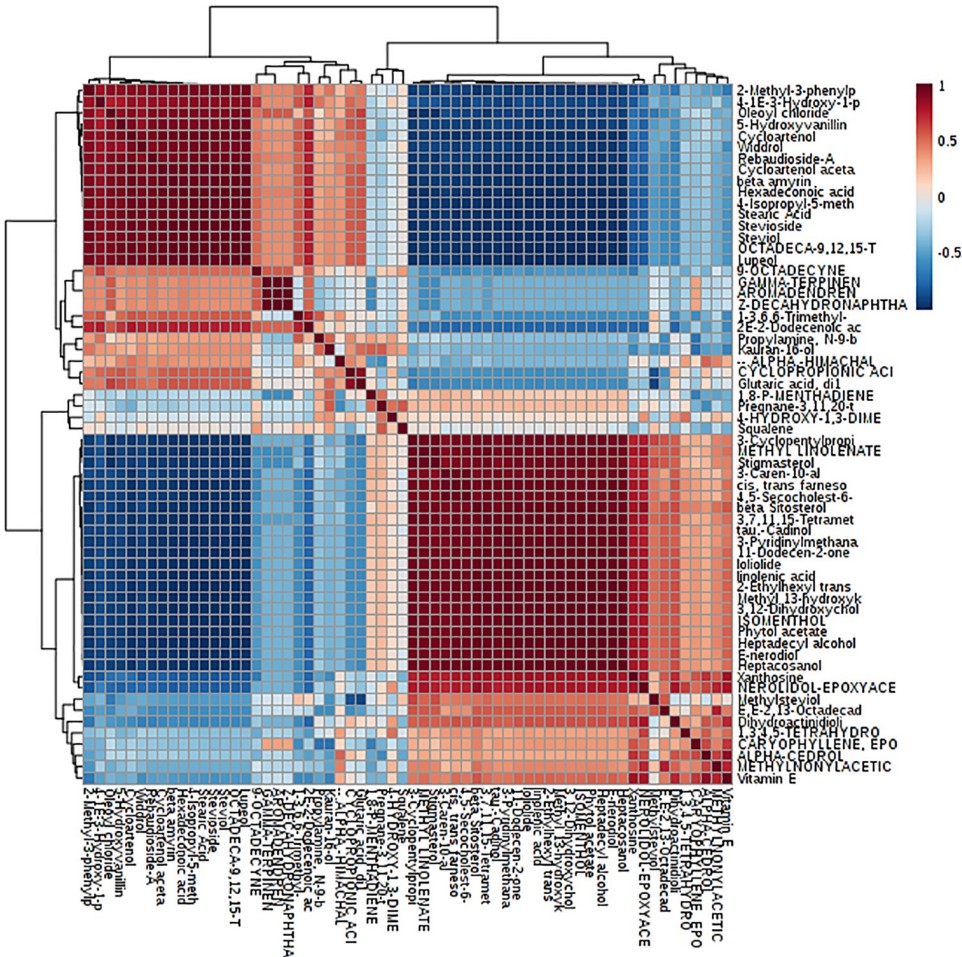

**Fig 16. Correlation table shows the high correlation between steviol and rebaudioside A.**

rebaudioside-A to stevioside ratio increased from 0.74 (control) to 2.83. In overall all the lines, RebA showed a positive correlation with steviol and stevioside. Taken together, our results suggest that with co-overexpression of *SrUGT6G1* and *SrKO* genes ultimately increased RebA concentration in stevia leaves.

In conclusion, concentration of RebA increased significantly with co- overexpression of *SrUGT6G1* and *SrKO* genes. Lines with increased RebA are more palatable and commercially viable.

## Supporting information

**S1 Raw images.**
(PDF)

## Acknowledgments

We would like to acknowledge UGC-SAP (DRS-I and DRS-II) for providing us the funds and infrastructure for conducting this research.

## Author Contributions

**Conceptualization:** Nazima Nasrullah, Monica Saifi, Malik Zainul Abdin.

**Data curation:** Nazima Nasrullah, Javed Ahmad, Syed Naved Quadri, Malik Zainul Abdin.

**Formal analysis:** Umara Nissar, Malik Zainul Abdin.

**Funding acquisition:** Malik Zainul Abdin.

**Investigation:** Irum Gul Shah, Malik Zainul Abdin.

**Methodology:** Nazima Nasrullah, Javed Ahmad, Monica Saifi, Syed Naved Quadri.

**Project administration:** Malik Zainul Abdin.

**Resources:** Malik Zainul Abdin.

**Software:** Nazima Nasrullah.

**Supervision:** Malik Zainul Abdin.

**Validation:** Nazima Nasrullah, Malik Zainul Abdin.

**Visualization:** Nazima Nasrullah, Monica Saifi, Irum Gul Shah, Umara Nissar, Syed Naved Quadri, Malik Zainul Abdin.

**Writing – original draft:** Nazima Nasrullah, Monica Saifi.

**Writing – review & editing:** Nazima Nasrullah, Monica Saifi, Irum Gul Shah, Umara Nissar, Syed Naved Quadri, Kudsiya Ashrafi, Malik Zainul Abdin.

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
