## [Decision Letter · Decision Letter 0]

25 May 2021

PONE-D-21-11562

Enhancement of diterpenoid steviol glycosides by co-overexpressing SrKO and SrUGT76G1 genes in <stevia rebaudiana=""> Bertoni.

PLOS ONE

Dear Dr. Nasrullah,

Thank you for submitting your manuscript to PLOS ONE. After careful consideration, we feel that it has merit but does not fully meet PLOS ONE’s publication criteria as it currently stands. Therefore, we invite you to submit a revised version of the manuscript that addresses the points raised during the review process.</stevia>

We look forward to receiving your revised manuscript.

Kind regards,

Allah Bakhsh

Academic Editor

PLOS ONE

Journal Requirements:

3. Please ensure that you include a title page within your main document. We do appreciate that you have a title page document uploaded as a separate file, however, as per our author guidelines (http://journals.plos.org/plosone/s/submission-guidelines#loc-title-page) we do require this to be part of the manuscript file itself and not uploaded separately.

4.We suggest you thoroughly copyedit your manuscript for language usage, spelling, and grammar. If you do not know anyone who can help you do this, you may wish to consider employing a professional scientific editing service.  

5. Please amend your manuscript to include your abstract after the title page.

6. Please upload a copy of Figure 13, to which you refer in your text on page 26. If the figure is no longer to be included as part of the submission please remove all reference to it within the text.

Additional Editor Comments:

Dear authors, I have now received comments on your manuscripts. You will see that the reviewers are suggesting changes to be incorporated in revised version before any further step. Therefore you are requested to address these changes in the form a rebuttal letter.

Reviewers' comments:

Reviewer's Responses to Questions

**Comments to the Author**

1. Is the manuscript technically sound, and do the data support the conclusions?

Reviewer #1: Yes

Reviewer #2: Yes

2. Has the statistical analysis been performed appropriately and rigorously? 

Reviewer #1: Yes

Reviewer #2: Yes

3. Have the authors made all data underlying the findings in their manuscript fully available?

Reviewer #1: No

Reviewer #2: No

4. Is the manuscript presented in an intelligible fashion and written in standard English?

Reviewer #1: Yes

Reviewer #2: No

5. Review Comments to the Author

Reviewer #1: The authors Nasrullah et al. presented their research entitled “Enhancement of diterpenoid steviol glycosides by co-overexpressing SrKO and SrUGT76G1 genes in Stevia rebaudiana Bertoni”. They have demonstrated the functional characterization of SrUGT76G1 and SrKO by over-expression in stevia via transient or stable transformation. The study is well designed and performed. However, there are some deficiencies which should be addressed to disseminate the detailed information to the scientific community.

Materials and Methods

1. In Materials and Methods headings 2.4 and 2.5 Transient transformation has been repeated twice. It should be incorporated into single heading.

2. For the Heading 2.6 Screening, in results section there is no figure of GFP in plant tissues?

3. Eight commonly used reference genes (RGs) (SrβTUB, SrARF1, SrαTUB, SrUBQ10, SrSAND, SrGAPDH, SrEF-1α, and SrACT7) were selected and their sequences were obtained from the TAIR database (http://www.arabidopsis.org). How the genes of stevia were taken from Arabidopsis database?

Results

1. In the results section “3.3 Molecular analysis of transformed and non-transformed stevia plants” the figure 3 which is a gel picture, is hard to understand. Arrows with PCR product size should be given. One gel picture contains 2 types of PCR products. The figures should be assigned with Lettering.

2. “3.5.9 Analysis of metabolites” It would be better if figure 9 presented as Fig. 9 A, Fig. B and Fig. 9C for the clarity of the results.

3. Under the heading of “3.5.10 Metabolic changes and correlation of metabolite levels” Figure 13 is mentioned. However, there is no figure in the manuscript.

In addition, there are some English grammar deficiencies those should be improved.

Reviewer #2: Introduction:

Introduction part was well written but still contains mistakes related to English grammar. Improve them. Authors should give page number and line number to ease the process of revision, so that errors/mistakes can be highlighted corresponding to specific lines. Some general comments in the introduction part are as follows:

• Authors must define the acronyms MVA and MEP pathways, at their first mention in the text such as Mevalonate and Methylerythritol Phosphate Pathways. (3rd paragraph of introduction)

• Likewise, please mention the acronym KO. The step in GA biosynthesis is catalyzed by ent-kaurene oxidase (KO) leading to the synthesis of…………….. (3rd paragraph of introduction).

Materials and Methods:

Materials and methods section was well elaborated with scientific reasonings; however, it needs some minor improvements as indicated below:

• In subsection 2.1. Chemicals for molecular research were obtained or purchased from….. (Use scientific style of writing).

• In sub section 2.2. Rephrase this sentence “The plants were grown in plastic pots (with a diameter of 31 and a height of 27 cm) in 1:1 ratio of soil and sand in poly house at Jamia Hamdard, New Delhi, India under controlled conditions, in an artificial climate chamber programmed for at 25°C/16°C 16 h/8 h day/night environment with a light intensity of 300 μmol/m2s-1 and relative humidity, 60%”

• Space should be given between a numerical value and unit like 100 mg. Please follow the Journal style for further rectification.

• GFP should be in capital letters. Please replace this in the complete manuscript.

• Why 2.4 and 2.5 subsections shared same heading “transient transformation”??

• It is mentioned in the text 1:1 ratio, which is wrong. The symbol “ : ” represents ratio, therefore no need to say 1:1 ratio. Remove the word ratio in the text.

Results:

This section needs improvements in following aspects as listed below:

• CAMVS35/SrKO ?? Is it CAMVS35 or CAMV 35S? In figure the authors have mentioned CAMV 35S whereas in text it is written CAMVS35. Please use the similar and standard format in the whole manuscript to avoid confusion (see section 3.1 and 3.3 etc..)

• Authors claimed that “The transformed leaves were confirmed by visualising the leaves under Leica TCS SP2 confocal microscope equipped with long-lasting solid-state lasers (Leica Microsystems, Germany), BUT NO GFP visualization of transiently expressed plant tissues has been given in the form of images in the manuscript? Please elaborate?? If GFP has been visualized through microscope, it must follow by proper images. Authors must give present appropriate images of GFP visualization (sec 3.1)

• Principal component analysis (PCA) of the stevioside and rebaudioside-A gave a clear cluster pattern (Tag this line with a relevant figure?) (sec 3.1)

• Relevant figures and tables should be mentioned and tagged in section 3.1. The results of increase and decrease of quantification studies and co expression results have been discussed without mentioning pertinent figures.

• Section 3.4 gave most of the details about How the morphological analysis of shoots and leaves were carried out. This is specifically a part of materials and methods section. Please focus on the results obtained in this section.

• The result presented in Fig. 5 does not correlate with these lines in the text (In the transformed leaf, SrUBQ10 has a minimum average Cq value of 15.62, while SrSAND has a maximum average Cq value of 22.83). Please check it again and took help of statistician.

• Fig.13: Correlation table shows the high correlation between steviol and rebaudioside A is not present in the manuscript.

Discussion:

In this section, please give more reasonings about why the co-overexpression of both genes resulted in decrease of stevioside content, whereas increases RebA amounts in comparison to individual expression studies. Since, no change in RebA to stevioside ratio was observed in overexpression of SrKO. Support your results by giving reasonings and explanations from the biosynthetic pathways of both genes involved or their homologs in other plants.

General Reviewers Comments:

Improve English grammar and sentence structure of the manuscript. Make all necessary corrections as indicated above in each section of the manuscript. Keeping in view the significance of stevia plant and originality of research conducted by the researchers, the study holds significance and is worthy enough to be accepted in the Journal after a major revision.

6. PLOS authors have the option to publish the peer review history of their article (what does this mean?). If published, this will include your full peer review and any attached files.

Reviewer #1: No

Reviewer #2: No

---

## [Author Response · Author response to Decision Letter 0]

7 Sep 2021

Dear Editor, 

Ploseone

Thank you for giving us the opportunity to submit a revised draft of our manuscript titled 

“Enhancement of diterpenoid steviol glycosides by co-overexpressing SrKO and SrUGT76G1 genes in Stevia rebaudiana Bertoni. Manuscript Number: PONE-D-21-11562”. We truly appreciate the time and effort that you and reviewers have dedicated to providing your valuable feedback on our manuscript. We are grateful to the reviewers for their insightful comments on our paper. We have incorporated changes suggested by the reviewers. We have highlighted the changes in the manuscript. 

Here are the point to point response to the reviewers’ comments and concerns.

Reviewers Comments

Introduction:

Introduction part was well written but still contains mistakes related to English grammar. Improve them. Authors should give page number and line number to ease the process of revision, so that errors/mistakes can be highlighted corresponding to specific lines. Some general comments in the introduction part are as follows:

Response: As suggested english grammar has been checked by experts and by online available software Premium ProwritingAid. 

• Authors must define the acronyms MVA and MEP pathways, at their first mention in the text such as Mevalonate and Methylerythritol Phosphate Pathways. (3rd paragraph of introduction)

Response: Acronyms have been properly defined in the manuscript in page 1, line 17.

• Likewise, please mention the acronym KO. The step in GA biosynthesis is catalyzed by ent-kaurene oxidase (KO) leading to the synthesis of…………….. (3rd paragraph of introduction).

Response: Acronyms have been properly defined in the manuscript in page 1, line 19.

Materials and Methods:

Materials and methods section was well elaborated with scientific reasonings; however, it needs some minor improvements as indicated below:

• In subsection 2.1. Chemicals for molecular research were obtained or purchased from….. (Use scientific style of writing).

 Response: We have changed the sentence in a more scientific style as suggested on page 2, line 56-57.

• In sub section 2.2. Rephrase this sentence “The plants were grown in plastic pots (with a diameter of 31 and a height of 27 cm) in 1:1 ratio of soil and sand in poly house at Jamia Hamdard, New Delhi, India under controlled conditions, in an artificial climate chamber programmed for at 25°C/16°C 16 h/8 h day/night environment with a light intensity of 300 μmol/m2s-1 and relative humidity, 60%”

Response: We have rephrased the sentence in a more scientific style as suggested on page 3, line 73-75.

• Space should be given between a numerical value and unit like 100 mg. Please follow the Journal style for further rectification.

Response: Space has been incorporated as suggested in page 3, line 77.

• GFP should be in capital letters. Please replace this in the complete manuscript. 

Response: We have replaced gfp by GFP at all places in the manuscript.

• Why 2.4 and 2.5 subsections shared same heading “transient transformation”??

Response: Thank you for pointing this out, as suggested we have changed the name of the subsection 2.4.

• It is mentioned in the text 1:1 ratio, which is wrong. The symbol “ : ” represents ratio, therefore no need to say 1:1 ratio. Remove the word ratio in the text.

Response: we have corrected it as suggested.

Results:

This section needs improvements in following aspects as listed below:

• CAMVS35/SrKO ?? Is it CAMVS35 or CAMV 35S? In figure the authors have mentioned CAMV 35S whereas in text it is written CAMVS35. Please use the similar and standard format in the whole manuscript to avoid confusion (see section 3.1 and 3.3 etc..)

Response: We have corrected it as suggested.

• Authors claimed that “The transformed leaves were confirmed by visualising the leaves under Leica TCS SP2 confocal microscope equipped with long-lasting solid-state lasers (Leica Microsystems, Germany), BUT NO GFP visualization of transiently expressed plant tissues has been given in the form of images in the manuscript? Please elaborate?? If GFP has been visualized through microscope, it must follow by proper images. Authors must give present appropriate images of GFP visualization (sec 3.1)

Response: We have incorporate the Figures of GFP visualization in the manuscript.

• Principal component analysis (PCA) of the stevioside and rebaudioside-A gave a clear cluster pattern (Tag this line with a relevant figure?) (sec 3.1)

Response: Thank you for pointing this sentence out, the sentence was mistakenly incorporated here and has been removed.

• Relevant figures and tables should be mentioned and tagged in section 3.1. The results of increase and decrease of quantification studies and co expression results have been discussed without mentioning pertinent figures.

Response: Figures have been incorporated as suggested on page 8, line 223.

• Section 3.4 gave most of the details about How the morphological analysis of shoots and leaves were carried out. This is specifically a part of materials and methods section. Please focus on the results obtained in this section.

Response: Modified as suggested at page 9, line 272.

• The result presented in Fig. 5 does not correlate with these lines in the text (In the transformed leaf, SrUBQ10 has a minimum average Cq value of 15.62, while SrSAND has a maximum average Cq value of 22.83). Please check it again and took help of statistician. 

Response: Cq value is the average of all the samples.

• Fig.13: Correlation table shows the high correlation between steviol and rebaudioside A is not present in the manuscript.

Response: Correlation table has been incorporated in the manuscript.

Discussion:

In this section, please give more reasonings about why the co-overexpression of both genes resulted in decrease of stevioside content, whereas increases RebA amounts in comparison to individual expression studies. Since, no change in RebA to stevioside ratio was observed in overexpression of SrKO. Support your results by giving reasonings and explanations from the biosynthetic pathways of both genes involved or their homologs in other plants. 

Response: Reasoning has been given in the manuscript as suggested on page 19, line 502 and 510.

General Reviewers Comments:

Improve English grammar and sentence structure of the manuscript. Make all necessary corrections as indicated above in each section of the manuscript. Keeping in view the significance of stevia plant and originality of research conducted by the researchers, the study holds significance and is worthy enough to be accepted in the Journal after a major revision.

Response: We have made all the changes as suggested, we look forward to hearing from you in due time regarding our submission and to respond to any further questions and comments you may have.

Sincerely,

Dr. Nazima Nasrullah

---

## [Decision Letter · Decision Letter 1]

3 Nov 2021

Enhancement of diterpenoid steviol glycosides by co-overexpressing SrKO and SrUGT76G1 genes in <stevia rebaudiana=""> Bertoni.

PONE-D-21-11562R1</stevia>

Dear Dr. Nasrullah,

We’re pleased to inform you that your manuscript has been judged scientifically suitable for publication and will be formally accepted for publication once it meets all outstanding technical requirements.

Kind regards,

Allah Bakhsh

Academic Editor

PLOS ONE

Additional Editor Comments (optional):

Reviewers' comments:

Reviewer's Responses to Questions

**Comments to the Author**

1. If the authors have adequately addressed your comments raised in a previous round of review and you feel that this manuscript is now acceptable for publication, you may indicate that here to bypass the “Comments to the Author” section, enter your conflict of interest statement in the “Confidential to Editor” section, and submit your "Accept" recommendation.

Reviewer #1: All comments have been addressed

Reviewer #2: All comments have been addressed

2. Is the manuscript technically sound, and do the data support the conclusions?

Reviewer #1: Yes

Reviewer #2: Yes

3. Has the statistical analysis been performed appropriately and rigorously? 

Reviewer #1: Yes

Reviewer #2: Yes

4. Have the authors made all data underlying the findings in their manuscript fully available?

Reviewer #1: Yes

Reviewer #2: Yes

5. Is the manuscript presented in an intelligible fashion and written in standard English?

Reviewer #1: Yes

Reviewer #2: Yes

6. Review Comments to the Author

Reviewer #1: (No Response)

Reviewer #2: (No Response)

7. PLOS authors have the option to publish the peer review history of their article (what does this mean?). If published, this will include your full peer review and any attached files.

Reviewer #1: No

Reviewer #2: **Yes: **Dr. Muhammad Naeem

---

## [Editor Report · Acceptance letter]

12 Jan 2022

PONE-D-21-11562R1 

Enhancement of diterpenoid steviol glycosides by co-overexpressing *SrKO* and *SrUGT76G1* genes in *Stevia rebaudiana Bertoni.*

Dear Dr. Nasrullah:

I'm pleased to inform you that your manuscript has been deemed suitable for publication in PLOS ONE. Congratulations! Your manuscript is now with our production department. 

Kind regards, 

on behalf of

Dr. Allah Bakhsh 

Academic Editor

PLOS ONE